# Application of the Doppler Spectrum of the Backscattering Microwave Signal for Monitoring of Ice Cover: A Theoretical View

Vladimir Karaev *, Yury Titchenko, Maria Panfilova, Maria Ryabkova, Eugeny Meshkov and Kirill Ponur

Federal Research Center Institute of Applied Physics of the Russian Academy of Sciences (IAP RAS), 603950 Nizhny Novgorod, Russia; yuriy@ipfran.ru (Y.T.); marygo@mail.ru (M.P.); m.rjabkova@gmail.com (M.R.); meshkovevgeny@gmail.com (E.M.); ponur@ipfran.ru (K.P.)
* Correspondence: volody@ipfran.ru; Tel.: +7-902-304-86-02

**Abstract:** In the radar remote sensing of sea ice, the main informative parameter is the backscattering radar cross section (RCS), which does not always make it possible to unambiguously determine the kind of scattering surface (ice/sea waves) and therefore leads to errors in estimating the area of the ice cover. This paper provides a discussion of the possibility of using the Doppler spectrum of the reflected microwave signal to solve this problem. For the first time, a semi-empirical model of the Doppler spectrum of a radar microwave signal reflected by an ice cover was developed for a radar with a wide antenna beam mounted on a moving carrier at small incidence angles of electromagnetic waves (0°–19°). To describe the Doppler spectrum of the reflected microwave signal, the following parameters were used: shift and width of the Doppler spectrum, as well as skewness and kurtosis coefficients. Research was conducted on the influence of the main parameters of the measurement scheme (movement velocity, width of antenna beam, sounding direction, incidence angle) and the sea ice concentration (SIC) on the parameters of the Doppler spectrum. It was shown that, in order to determine the kind of scattering surface, it is necessary to use a wide or knife-like (by the incidence angle) antenna. Calculations confirmed the assumption that, when measured from a moving carrier, the Doppler spectrum is a reliable indicator of the transition from one kind of scattering surface to another. The advantage of using the coefficients of skewness and kurtosis in the analysis is that it is not necessary to keep the radar velocity unchanged during the measurement process.

**Keywords:** Doppler spectrum of the reflected microwave signal; ice cover; sea waves; antenna beam; width and shift of the Doppler spectrum; skewness and kurtosis coefficients; sea ice concentration

## 1. Introduction

Climate change is one of the most serious threats to the future of humanity. Although there is no generally accepted scenario of global warming, observed variants of climate change are observed to be unfavorable, for example, [1,2].

The natural phenomenon most sensitive to climate change is the ice cover in the Arctic and Antarctic, which allows us to consider it as an indicator of the warming process. One of the characteristics of the ice cover is its area. An example of visualization of the dynamics of the ice cover in the Arctic from 1984 to 2019 is given on the website [3]. The seasonal and interannual variability of the ice area is clearly visible.

To take measurements of ice cover, radiometers, radars with real and synthetic aperture, and optical and infrared sensors can be installed on aerospace carriers, for example, [4–12].

In radar remote sensing of the ice cover, the main information parameter is the backscattering radar cross section (RCS). As an example, Figure 1 shows the dependence of the RCS on the incidence angle for ice cover (Figure 1a) and sea waves (Figure 1b), plotted according to the ASCAT scatterometer data [13].

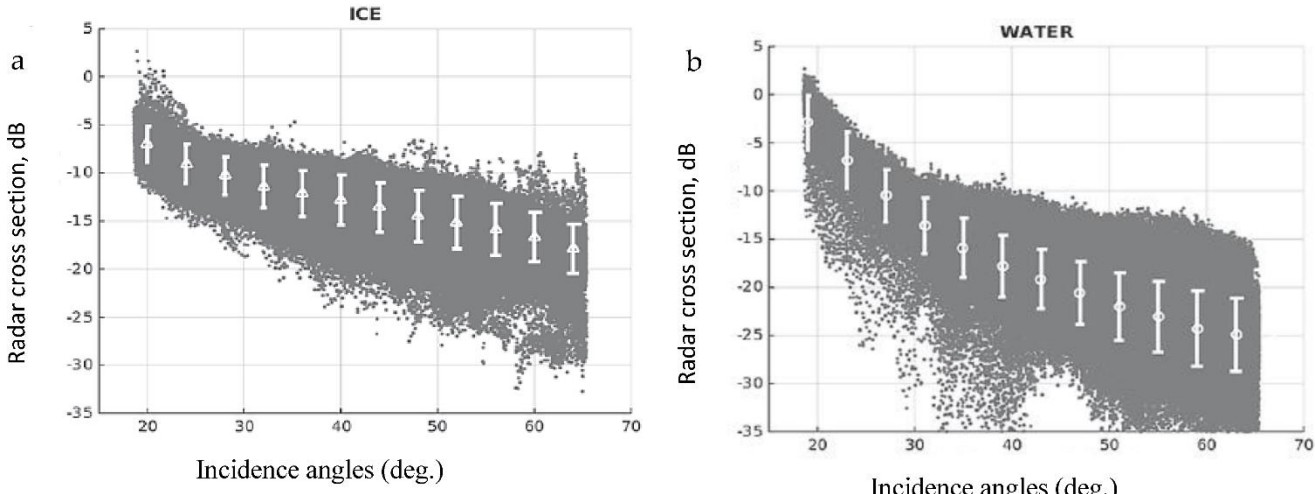

**Figure 1.** Dependence of the RCS on the incidence angle for ice cover (**a**) and sea waves (**b**) [13].

The scatter of the RCS over the open water surface (Figure 1b—water) and ice cover (Figure 1a—ice) observed in the figure is due to many factors, for example, wind speed (Figure 1b) and type of ice: nilas, pancake, gray ice, etc. (Figure 1a). With such a spread in the values of the RCS, it is impossible to determine the kind of scattering surface from single measurements, since there is an ambiguous relationship between the RCS and the kind of the underlying surface.

The average values in the figure are shown as white dots and the bar shows the dispersion. There is a slight difference in the angular dependence of the sea waves and ice, so it becomes possible to separate the ice cover and sea waves according to average dependences. However, even for averaged values, the result of classifying the kind of scattering surface is ambiguous but of a probabilistic nature.

Thus, the RCS is not the optimal parameter for the problem of determining the kind of the underlying surface.

Therefore, at middle incidence angles, the determination of the kind of scattering surface (ice/water) may exist for the averaged data (white circles within the "cloud" of data) and the solution becomes ambiguous for non-averaged values.

The variability of sea waves and different types of sea ice leads to a spread in the values of the RCS, which is clearly seen in Figure 2 [8]. The measurements were made by X-band radar (3 cm) at an incidence angle of 40° and in vertical polarization (VV).

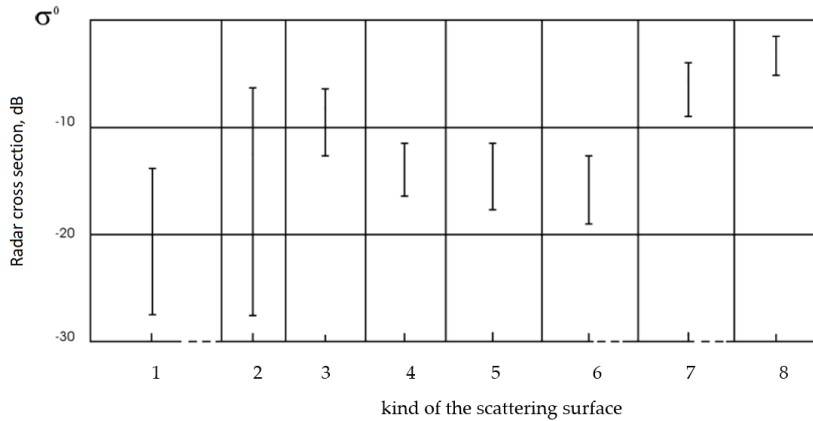

**Figure 2.** Generalized dependence of the sea ice RCS (VV-polarization, incidence angle 40°): 1—sea waves at wind speeds up to 10 m/s; 2—nilas, ice thickness less than 10 cm; 3—young ice, thickness 10–30 cm; 4—thin first-year ice, thickness 30–70 cm; 5—average first-year ice, thickness 70–120 cm; 6—thick first-year ice, thickness > 120 cm; 7—multi-year ice, thickness > 200 cm; 8—ice shelves [8].

In the Figure 2, the following kinds of scattering surfaces are presented: 1—sea waves at wind speeds up to 10 m/s; 2—nilas, ice thickness less than 10 cm; 3—young ice, thickness 10–30 cm; 4—thin first-year ice, thickness 30–70 cm; 5—average first-year ice, thickness 70–120 cm; 6—thick first-year ice, thickness > 120 cm; 7—multi-year ice, thickness > 200 cm; 8—ice shelves [8].

The observed scatter makes it difficult to solve the problem of determining the kind of underlying surface.

When electromagnetic waves are reflected, not only the energy but also the spectral characteristics of the radar signal change, which probably have not been analyzed in terms of ice cover. We are not aware of any papers in which the model of the Doppler spectrum over the ice cover was considered.

It is known that Doppler radar (HF or microwave radar) may be used for measurements of the movement velocity of the icebergs or floes. Measurement of sea ice velocity was demonstrated using an HF radar [14,15]. Detailed research of iceberg detection was presented in the report [16].

Ground-based microwave radars, such as C-band [17] and X-band radars [18], can provide high spatial and temporal resolutions in real time, but because microwave propagation is limited to line of sight, these radars must be installed on high mountains or buildings for long-range observations.

An interesting project was realized in Japan. The Institute of Low Temperature Science, Hokkaido University, Japan, has operated radar systems on the northern coast of Hokkaido to monitor the coastal sea zone. In 1969, an operative C-band sea-ice radar (SIR) network was established for continuous monitoring of the sea-ice conditions [17]. The radars operated at C-band (wavelength 5.4 cm) at large incidence angles. Due to high spatial resolution (1.5 km), radar can detect the position of ice edge. The SIR system provides information on ice concentration, ice kinematics, and ice-edge dynamics, which is required by the local communities along the northern coast of Hokkaido. However, this system was replaced by HF radars in 2004.

This study focuses on the development of a Doppler spectrum model for the ice cover and analyzing the properties of the Doppler spectrum in terms of developing algorithms for classifying the scattering surface according to the ice/water criterion.

## 2. Method

### 2.1. Initial Assumptions

We will consider the area of small incidence angles, when, in the case of the sea waves, the quasi-specular backscattering mechanism is dominant, and the Kirchhoff method is used to find the reflected signal, for example, [19–22]. It is obvious that for the ice cover, the concept of the Doppler spectrum exists only when measured from a moving carrier. In previous studies, it was shown that in the case of a fast-moving carrier, the Doppler spectrum width depends on the parameters of sea waves only for a radar with a wide antenna beam [23]; for a narrow antenna beam, the Doppler spectrum width depends only on velocity of movement. The calculations were performed for an orbital radar (velocity 7000 m/s), so we will repeat them for the aircraft version, when the flight speed is less than, for example, 200 m/s.

Figure 3 shows the dependence of the Doppler spectrum width on the antenna beam width for two wind speeds (5 and 10 m/s, fully developed wind waves) for a moving carrier (velocity 200 m/s along the axis $Y$, probing direction 45°, radar wavelength 0.021 m, and an incidence angle 5°). The used probing scheme is shown in Figure 4.

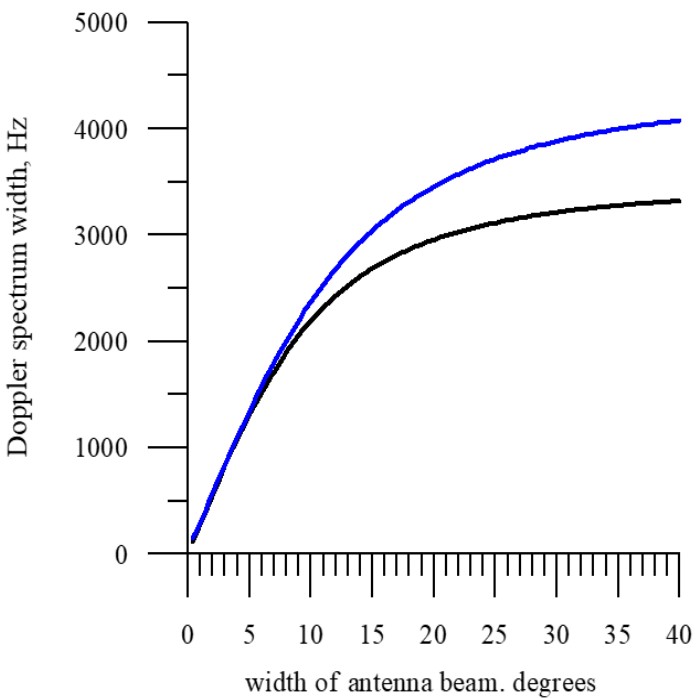

**Figure 3.** Dependence of the Doppler spectrum width on the antenna beam width for a wind speed of 5 m/s (black curve) and 10 m/s (blue curve) in the case of a fully developed wind wave. The carrier velocity is 200 m/s along the axis *Y*, the incidence angle is 5°, and the probing direction is 45°. The radar wavelength is 0.021 m.

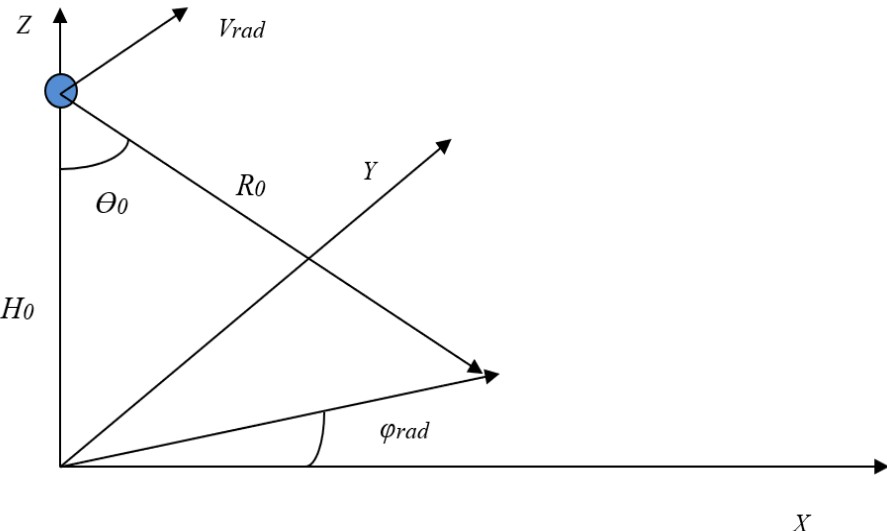

**Figure 4.** Measurement scheme.

It can be seen from Figure 3 that despite the fact that the wave parameters for different wind speeds are very different, in the case of a narrow antenna beam, this does not affect the width of the Doppler spectrum, and the curves coincide. Therefore, for a narrow antenna beam, the main factor affecting the width of the Doppler spectrum is the velocity of radar movement.

With an increase in the width of the antenna beam, the dependences of the Doppler spectrum width for different sea wave intensities (in this case, wind speeds) are separated. Thus, a radar with a wide antenna beam begins to "see" the reflecting surface, and the width of the Doppler spectrum depends not only on the velocity of movement but also on the parameters of sea waves.

From the theoretical model of the Doppler spectrum, it follows that for a radar with a wide antenna beam, with an increase in the movement velocity of radar, the key factor is not the orbital velocities (movement of the reflecting surface) but rather the mean square slopes (*mss*) of large-scale waves compared to the radar wavelength [23,24].

If we make a number of simplifying assumptions about the direction of probing, the direction of carrier movement, and the direction of wave propagation, the formula for the width of the Doppler spectrum [24] can be greatly simplified and written in the following form:

$$\Delta F_{20} \sim V_{rad} \cos \theta_0 \sqrt{\frac{\delta_\alpha^2 mss_{yy}}{11.04 mss_{yy} + \delta_\alpha^2}} \tag{1}$$

where $\theta_0$ is the incidence angle; $mss_{yy}$ is the mean square slopes (*mss*) of large scale, in comparison with the radar wavelength, sea waves (large-scale waves) along axis $Y$; $V_{rad}$ is the velocity of radar movement; $\delta_\alpha$ is the width of the antenna beam at the level 0.5 on power.

It can be seen from the formula that if it is to use a narrow antenna beam ($\delta_\alpha^2 < mss_{yy}$), it is possible to neglect the $mss_{yy}$ of large-scale waves in the sum, and then the $mss_{yy}$ will be reduced. As a result, the width of the Doppler spectrum will be proportional to the width of the antenna beam.

Conversely, for a wide antenna beam, the fraction will be reduced in such a way that the Doppler spectrum width (see Formula (1)) will be proportional to the $mss_{yy}$ of large-scale waves. Therefore, a change in the *mss* of the reflecting surface leads to a change in the surface scattering diagram (dependence of the RCS on the angle of reflection), which ultimately affects the Doppler spectrum width when measured from a moving carrier.

The *mss* of the ice cover and sea waves are very different, so it was assumed that, when measuring from a moving carrier in terms of the width and shift of the Doppler spectrum, it would be easy to separate the ice cover and sea waves. This work is devoted to testing this assumption.

### 2.2. Semi-Empirical Model of the Doppler Spectrum for Ice Cover

For the sea surface, the description in terms of the wave spectrum is generally accepted, and many models of wave spectra are currently known, for example, [25–29]. Due to this, it is possible to obtain analytical formulas for the Doppler spectrum at small incidence angles, for example, [30–36].

To study the properties of the Doppler spectrum backscattered by the sea surface, numerical methods are used, for example, [37–41]. A wave spectrum of sea waves is used to model a scattering surface, so a spectral description is also required when using the standard ice cover modeling approach. There is no spectral description for the ice cover, so it is necessary to use another approach to develop a semi-empirical model of the Doppler spectrum, which will be based on the available experimental data.

The measurement scheme is shown in Figure 4. The radar is mounted on an aircraft that is moving at a velocity of $V_{rad}$ along the $Y$ axis at a height of $H_0$. The incidence angle is equal to $\theta_0$ and the sounding is carried out at an angle of $\varphi_{rad}$ in the $XY$ plane. The slant range to the reflection point is $R_0$. Then, the radial velocity component for the reflecting point is given by the following formula:

$$V_r = V_{rad} \sin \varphi_{rad} \sin \theta_0 \tag{2}$$

When measuring the Doppler spectrum from a moving carrier, the width of the antenna beam is important, determining the size of the reflecting area (footprint) and the spread of radial velocities in the reflected radar signal. In calculations, it was assumed that the antenna beam is Gaussian and is written in the following form:

$$G(\alpha, \beta) = \exp\left[-1.38\left(\frac{(\theta_0 - \alpha)^2}{\delta_\alpha^2} + \frac{(\varphi_{rad} - \beta)^2}{\delta_\beta^2}\right)\right] \tag{3}$$

where $\delta_\alpha$ and $\delta_\beta$ are the antenna beam width at a 0.5 power level; $\alpha$ and $\beta$ are the incidence angle and azimuth angle within the antenna beam, respectively, measured from the beam axis ($\theta_0$, $\varphi_{rad}$), i.e., $\theta = \theta_0 + \alpha$ and $\varphi = \varphi_{rad} + \beta$.

A change in the azimuth angle leads to a change in the incidence angle; therefore, to correctly calculate the radial velocity, it is necessary to recalculate the incidence angle using the following formula:

$$\theta_N = \text{arctg}\left(\frac{tg\theta}{\cos\beta}\right) \tag{4}$$

To find the Doppler spectrum of the backscattered signal, it is necessary to integrate over the scattering area:

$$S_{dop}(V_r) \sim \iint\limits_S G^4(\alpha,\beta)\,d\alpha\,d\beta \tag{5}$$

After integration, we obtained the spectrum of Doppler velocities (the distribution function of the radial velocity component). It is more common to represent the Doppler spectrum on the frequency axis; thus, to obtain the conventional Doppler spectrum, it is necessary to use the following formula:

$$f_r = \frac{2V_r(\alpha,\beta)}{\lambda} \tag{6}$$

where $\lambda$ is the radar wavelength.

Figure 5 shows examples of Doppler spectra for a moving carrier ($V_{rad}$ = 200 m/s), incidence angle $\theta_0$ = 5°, azimuth angle $\varphi_{rad}$ = 45°, and four values of the antenna beam: 2° × 2°, 2° × 20°, 20° × 2°, and 20° × 20°. For the convenience of comparison, we will always normalize each Doppler spectrum to its maximum. The first two spectra (2° × 2° and 2° × 20°) are shown in Figure 5a, and the last two Doppler spectra are shown in Figure 5b.

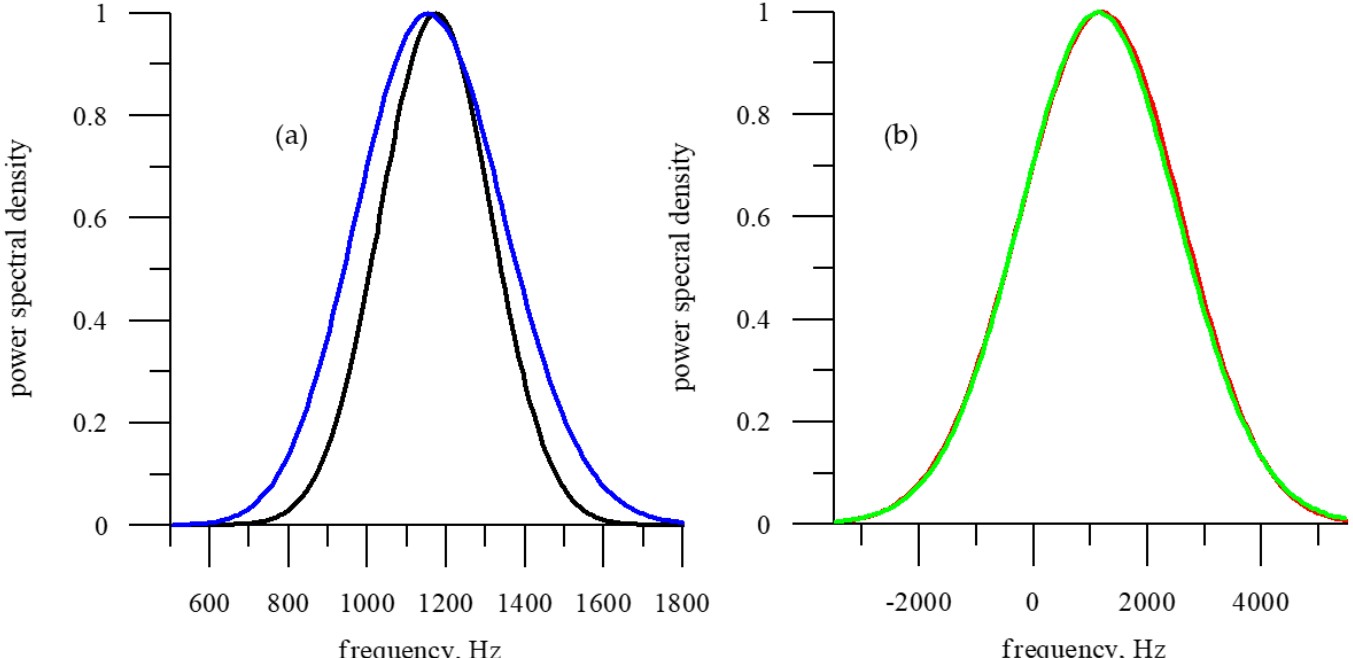

**Figure 5.** Normalized Doppler spectra for a moving carrier ($V_{rad}$ = 200 m/s), incidence angle $\theta_0$ = 5°, azimuth angle $\varphi_{rad}$ = 45°, and four values of the antenna beam: 2° × 2° (black curve), 2° × 20° (blue curve) (**a**), and 20° × 2° (red curve), 20° × 20° (green curve) (**b**).

When transitioning from a narrow antenna beam (2° × 2°—black curve) to a knife-like beam (2° × 20°—blue curve), due to the wide antenna beam in the azimuthal plane, the

range of incidence angles increases (see Formula (4)), which leads to a noticeable increase in the Doppler spectrum width. In this case, the width of the antenna beam in terms of the incidence angle is only 2°; therefore, a change in the azimuth angle $(+/-10°)$ provides a noticeable increase in the range of incidence angles. It leads to increasing the Doppler spectrum width.

This effect practically does not manifest itself when transitioning from a knife-like antenna $(20° \times 2°$—red curve) to a wide antenna $(20° \times 20°$—green curve). This is because, in contrast to the first case, the change in the incidence angle due to a change in the azimuth angle $(+/-10°)$ will be small compared to the width of the antenna beam along the incidence angle $(20°)$.

In calculations, it was assumed that all surface points have the same reflection coefficient, which is not true. Thus, the next step in developing a semi-empirical model of the Doppler spectrum is related to taking into account the scattering diagram of the ice cover (or the dependence of the RCS on the incidence angle).

In our research, we use Ku-band ($\lambda$ = 0.021 m) precipitation radar data from the TRMM (Tropical Rain Measuring Mission) and GPM (Global Precipitation Measurement) satellites [42,43].

Precipitation measurement is an important task, and a joint project between Japan and the United States was implemented to solve it. The TRMM (Tropical Rainfall Measuring Mission) satellite was the first precipitation satellite and was launched on 28 November 1997 from the Tanegashima Space Center (TNSC) (JAXA-TRMM). Precipitation radar (PR—Ku-band) on board the TRMM satellite measured the spatial distribution of rain in the tropical area. The TRMM satellite made observations for 17 years.

The dual-frequency precipitation radar (DPR) is a successor to the PR (13.6 GHz) loaded onto the GPM's (Global Precipitation Measurement) predecessor TRMM (JAXA-TRMM). The 35.5 GHz radar was additionally installed for high-accuracy observation of low-intensity rain. The launch of the core observatory for the GPM mission aboard was successfully performed on 28 February 2014. It can observe not only the tropical zone but also mid-to-high-latitude areas due to an orbit inclination of 65°.

DPR operates at wavelengths of 2.2 cm and 0.8 cm, and the probing scheme is shown in Figure 6. DPR and PR are designed to measure the spatial distribution of precipitation as well as its vertical profile to determine the precipitation intensity. The last resolution element contains data on backscattering from water or land surface.

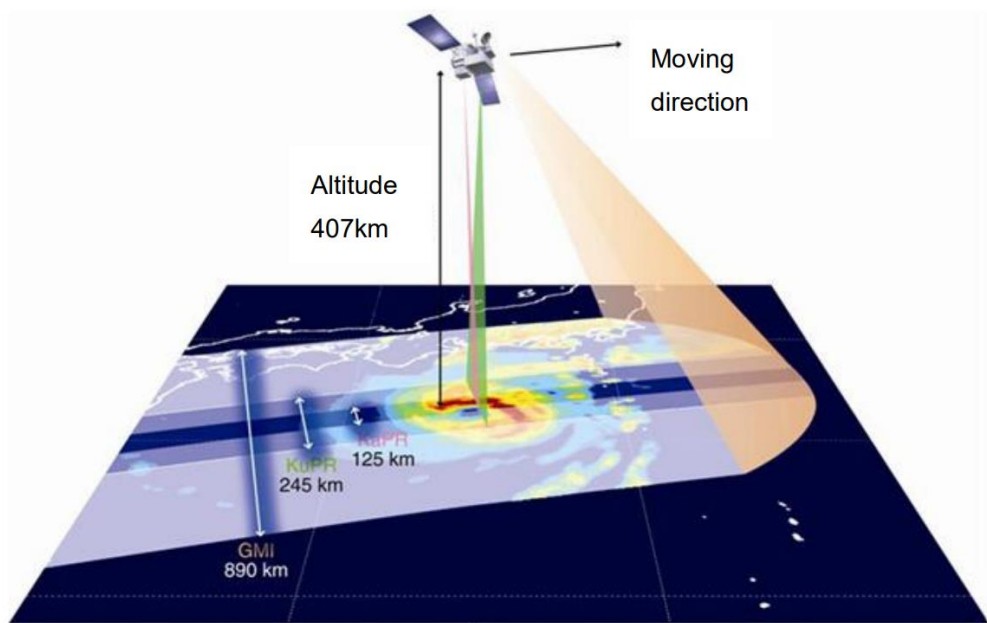

**Figure 6.** Scheme of measurements [43].

These data were used to determine the scattering diagram of the underlying surface.

In the Ku-band, measurements were taken for the range of incidence angles of 0°–19°. Precipitation radar data obtained over the Sea of Okhotsk were used to perform regression and derive formulas for an ice and sea surface backscatter diagram. An example of the dependence of the RCS on the incidence angle for a dry ice cover (negative air temperature, first-year ice) is shown in Figure 7 [44,45]. In the figure, stars of different colors represent different days.

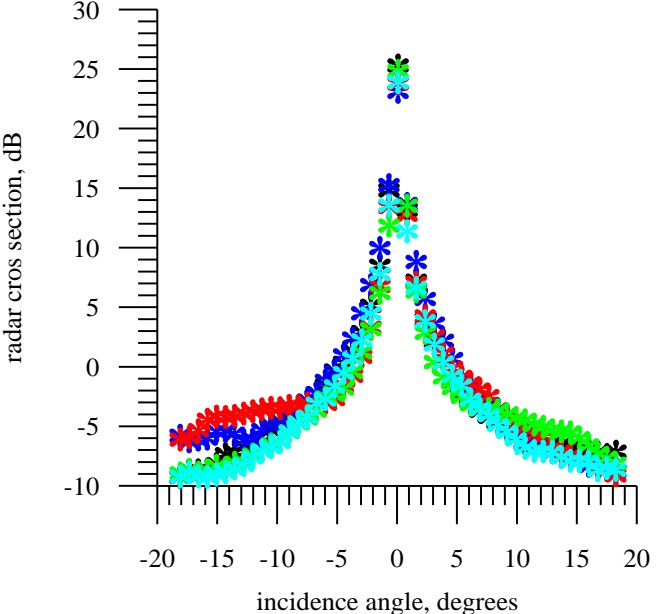

**Figure 7.** Dependence of the RCS on the incidence angle (Ku-band) for a "dry" ice cover (negative air temperature). Different colors represent different days.

A more extensive analysis of the dependence of the RCS on the incidence angle for ice cover and sea waves is undertaken in a paper currently under review [46] (private communication). However, using another dependency will not lead to fundamental changes in the results obtained, so we will use simpler formulas in the paper.

As a result of the regression analysis, the angular dependence of the backscatter diagram for the ice cover was approximated by the following formula:

$$RCS_{ice}(\theta) = a_{ice} + b_{ice}\theta + c_{ice}\theta^2 + d_{ice}\exp(-e_{ice}|\theta|) \tag{7}$$

where $a_{ice} = -3.1518$, $b_{ice} = -0.008708$, $c_{ice} = -0.016928$, $d_{ice} = 26.013$, $e_{ice} = 0.5288$.

Thus, to calculate the Doppler spectrum of the backscattered radar signal, it is necessary to integrate over the scattering area:

$$S_{dop}(V_r) \sim \iint\limits_{S} G^4(\alpha, \beta) \cdot RCS_{ice}(\theta)d\alpha d\beta \tag{8}$$

It should be noted that for the ice cover, the azimuthal dependence of the RCS (from a probing direction) can be neglected, since, in contrast to sea waves, the ice surface can be considered isotropic.

### 2.3. Semi-Empirical Model of the Doppler Spectrum for Sea Waves

For sea waves, there are analytical formulas for the Doppler spectrum of the backscattered radar signal; thus, the theoretical model can be used to assess the correctness of the proposed approach to the development of a semi-empirical model of the Doppler spectrum. To make such a comparison, the procedure needs to be repeated for developing

a semi-empirical model of the Doppler spectrum used for the ice cover, for sea waves. Let us define the dependence of the RCS on incidence angle in the following form:

$$RCS_{sea}(\theta) = a_{sea} + b_{sea}\theta + c_{sea}\theta^2 + d_{sea}\theta^3 + e_{sea}\theta^4 + f_{sea}\theta^5 \tag{9}$$

where $a_{sea}$ = 11.2912, $b_{sea}$ = 0.00626, $c_{sea}$ = −0.04076, $d_{sea}$ = −0.000104, $e_{sea}$ = 1.381 × 10$^{-5}$, and $f_{sea}$ = 7.911 × 10$^{-8}$. To find the dependence, we used measurements of a precipitation radar over the Sea of Okhotsk in the summer season, averaged over several days.

Figure 8 compares the model backscatter diagrams for ice cover (red curve, Formula (7)) and sea waves (black curve, Formula (9)). When plotting the figure, a transition was made to decibels. The figure shows that the behavior of the angular dependences for ice cover and sea waves is fundamentally different, and this property is used in the classification algorithm for the kind of scattering surface (ice/water) according to the RCS [47].

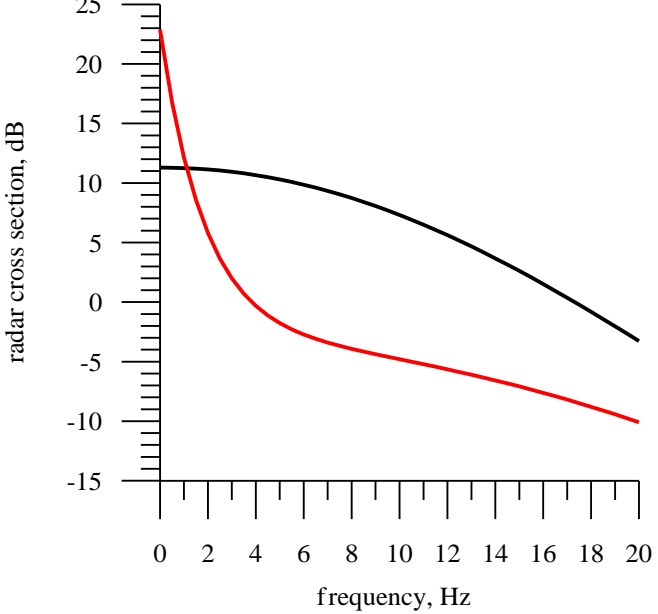

**Figure 8.** Dependence of the backscattering RCS on the incidence angle for ice cover (a red curve) and for sea waves (a black curve).

As in the case of middle incidence angles (Figure 1), the RCS section will strongly depend on the state of the scattering surface, for example, on the wind speed, but the shape of the dependence will not change, i.e., the second derivative will retain a sign. Note that the dependences of the RCS for the ice cover (Formula (7)) and the sea surface (Formula (9)) were obtained from a limited set of data and are not universal. They are used in this study to illustrate the difference in the angular dependences for ice and sea waves and to compare Doppler spectra measured over ice and sea waves.

### 2.4. Basic Parameters of Doppler Spectrum

Usually, two parameters are used to describe the Doppler spectrum: width $\Delta F_{20}$ and shift $f_{shift}$. For the measured Doppler spectrum $S_{dop}(f)$, the shift is calculated using the following formula:

$$f_{shift} = \frac{\int f \cdot S_{dop}(f)df}{\int S_{dop}(f)df} \tag{10}$$

There are several definitions for the width of the Doppler spectrum. In this work, we will use the following:

$$\Delta F_{20} = 2\sqrt{\frac{\int f^2 S_{dop}(f)df}{\int S_{dop}(f)df} - f_{shift}^2} \tag{11}$$

In addition, let us introduce one more definition of the Doppler spectrum width $\Delta F_{42}$, which is calculated in terms of the central statistical moments of the second order $\mu_2$ and the fourth order $\mu_4$:

$$\Delta F_{42} = \sqrt{\frac{\mu_4}{\mu_2}} = \sqrt{\frac{\mu_4}{\sigma^2}} \qquad (12)$$

where $\sigma^2$ is the dispersion of the process.

If the shape of the Doppler spectrum is close to Gaussian, two parameters are sufficient to describe it: width and shift. If the shape of the measured Doppler spectrum differs from Gauss, then the shape of the Doppler spectrum can be considered as an additional information parameter and must be used in the analysis of the reflected signal. Therefore, in order to make the description of the Doppler spectrum more complete, we considered two more characteristics: the kurtosis and skewness coefficients:

$$A = \frac{\mu_3}{\sigma^3} \text{ and } E = \frac{\mu_4}{\sigma^4} - 3 \qquad (13)$$

In other papers devoted to the Doppler spectrum, we did not find any mention of the use of skewness and kurtosis coefficients in relation to the Doppler spectrum. However, these are important characteristics of the Doppler spectrum, which provide new information, in particular, on the dominant backscattering mechanism, on sea currents [48,49].

*2.5. Comparison of Analytical and Semi-Empirical Models of Doppler Spectrum*

As noted earlier, for small incidence angles of probing radiation on the sea surface, there is a theoretical model of the Doppler spectrum, which was obtained in the Kirchhoff approximation. This will allow to evaluate the correctness of the method used to develop a semi-empirical model of the Doppler spectrum by comparing it with the theoretical model.

The theoretical model of the Doppler spectrum [24,31] includes the statistical characteristics of sea waves, which can be calculated from the wave spectrum model. The input parameters of the wave spectrum model are the wind speed and the nondimensional wind fetch [28]. Formula (9) describes the averaged dependence of the RCS on the incidence angle obtained from the DPR data. In the general case, the dependence of the RCS on the incidence angle is not unambiguous (see Figure 1), which complicates the problem of determining the kind of the scattering surface.

In the Kirchhoff approximation, the formula for the RCS for sea waves has the following form (Bass, Fuchs 1972) [19]:

$$\sigma_0(\theta) = \frac{\left| R_{eff}(0) \right|^2}{2\cos^4\theta \sqrt{mss_{xx}mss_{yy} - mss_{xy}^2}} \times \exp\left[ -\frac{\text{tg}^2\theta}{2\left( mss_{xx}mss_{yy} - mss_{xy}^2 \right)} \cdot mss_{yy} \right], \quad (14)$$

where $mss_{xx}$ and $mss_{yy}$ are the *mss* of large-scale waves along axis *X* and axis *Y*, respectively; $mss_{xy}$ is the non-normalized correlation coefficient between the slopes along the axes *X* and *Y* (hereinafter, the correlation coefficient); $R_{eff}$ is the effective reflection coefficient introduced to take into account the influence of a ripple on the power of the reflected signal.

Thus, the problem is reduced to determining the wind speed, which will give the best match between the model dependence from incidence angle (Formula (14)) and the experiment (Formula (9)). To achieve this, it is necessary to determine the wind speed that will provide the *mss* of large-scale waves observed in the experiment. The *mss* determines the form of the dependence of the RCS on the incidence angle (Figure 8), which makes it easy to estimate the accuracy of selecting the wind speed.

The performed analysis showed that if we consider sea waves propagating at an azimuthal angle of 45°, then for a fully developed wind wave, it is necessary to set the wind speed equal to 9.7 m/s. Based on the wave spectrum, the *mss* of large-scale waves were calculated and the result is shown in Figure 8: asterisks are obtained by Formula (9) and the red curve is plotted by Formula (14). That is, the form of the dependence of the

RCS on the incidence angle is important to us; therefore, for the convenience of comparison, the theoretical and experimental dependences were equated at a zero incidence angle when plotting the graph.

It can be seen from the Figure 9 that a good agreement between the theoretical dependence and the experiment was obtained; therefore, for further estimates, we will also assume that there is a fully developed wind wave on the surface, which was formed at a wind speed of 9.7 m/s and propagates at an azimuth angle of 45°.

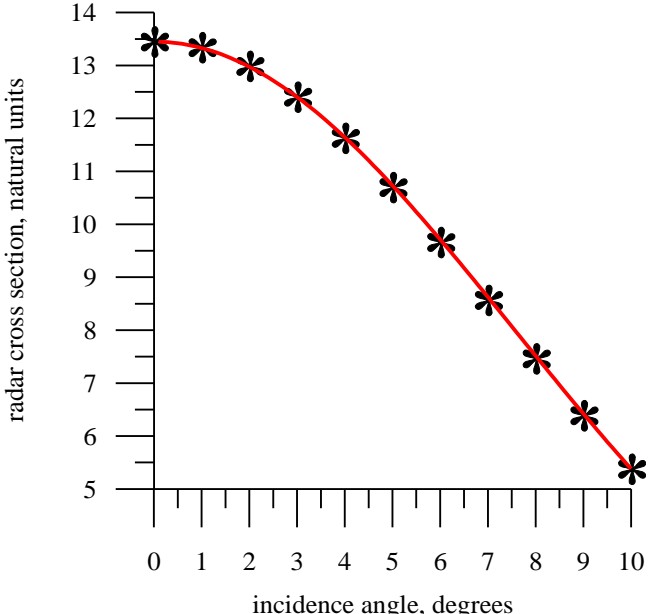

**Figure 9.** Dependence of the RCS on the incidence angle: the asterisks were obtained from the regression dependence (Formula (9)); the red curve is the result of calculation by Formula (14).

Wave spectrum [28] was used to calculate all statistical moments of the second order for a wind speed of 9.7 m/s and substituted into the theoretical formula for the Doppler spectrum [24]. Calculations were made for the following parameters: radar velocity of 200 m/s, incidence angle of 5°, and sounding direction of 45°.

For calculations, a radar with a knife-like antenna beam ($14° \times 2°$) was chosen. In Figure 10, the Doppler spectrum calculated from the semi-empirical model is shown with a green curve, and the theoretical model is shown with a dotted line. For ease of comparison, the spectra are normalized at their own maximum. It can be seen from the figure that for fast-moving radar, both models of the Doppler spectrum show close results, i.e., the proposed approach to developing a semi-empirical model of the Doppler spectrum is effective for a fast-moving radar.

Discrepancies between the theoretical model and the semi-empirical model may appear at low velocity because the velocity of the radar and the orbital velocities of the sea surface become comparable. This must be taken into account when making measurements and analyzing data. For example, Figure 11 shows the results of numerical simulation of the Doppler spectrum for a carrier velocity of 4 m/s (Figure 11a) and 20 m/s (Figure 11b). The blue curve in Figure 11a is built according to the theoretical model of the Doppler spectrum for a motionless radar ($V_{rad} = 0$). The direction of wave propagation is from the radar, so the Doppler spectrum shift is negative.

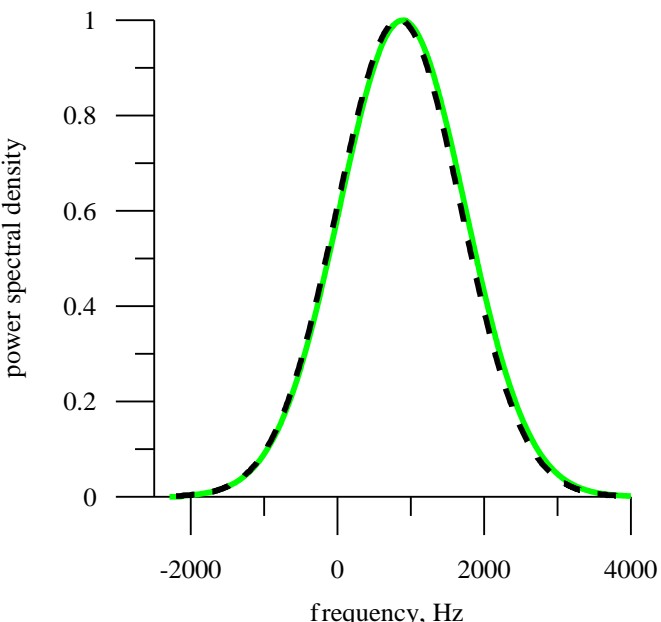

**Figure 10.** Normalized Doppler spectra for a moving carrier ($V_{rad}$ = 200 m/s), incidence angle $\theta_0 = 5°$, azimuth angle $\varphi_{rad} = 45°$, and antenna beam $14° \times 2°$: green curve—semi-empirical model; dotted line—theoretical model (wind speed 9.7 m/s).

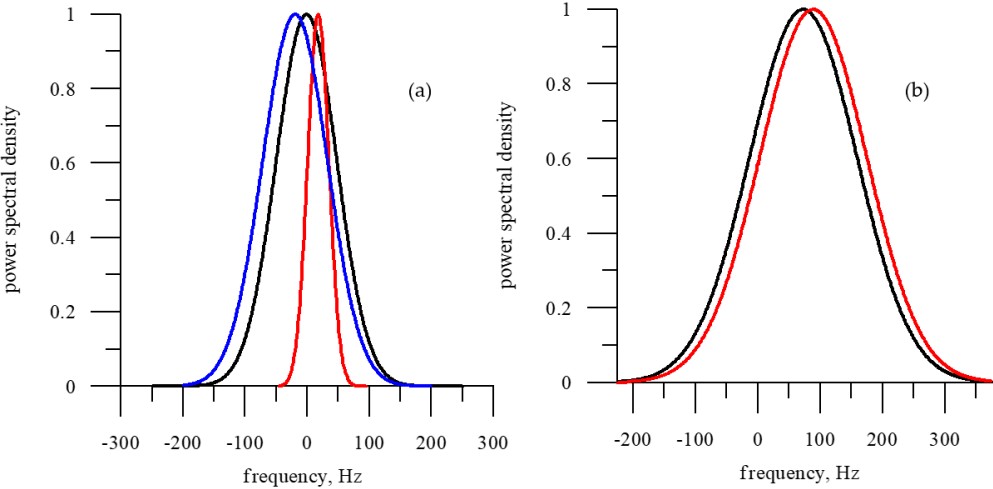

**Figure 11.** Examples of Doppler spectra at a radar velocity of 4 m/s (**a**) and 20 m/s (**b**). The blue curve is based on the theoretical model of the Doppler spectrum for the sea surface. The black curves are calculated using a semi-empirical formula for the sea surface. Doppler spectra for the stationary "sea" surface are shown as red curves.

The black curve is derived from a theoretical model of the Doppler spectrum for a radar velocity of 4 m/s and a sounding direction of 45°. A radar movement occurs along the $Y$ axis, which results in a positive Doppler shift and, as seen in Figure 11a, this almost cancels out the negative Doppler shift caused by sea waves.

The red curve was obtained for a radar moving over a stationary "sea" surface (Formula (8)). The surface is not moving, so the Doppler spectrum has narrowed considerably. In this case, the width is determined only by the movement of the radar, and the Doppler spectrum shift has a positive sign.

In Figure 11b, the calculations were made for a radar velocity of 20 m/s, the black curve was built using the theoretical model of the Doppler spectrum for sea waves, and the red curve was built using the semi-empirical model of the Doppler spectrum for a stationary

"sea" surface. It can be seen that, even at such a velocity, the widths of the Doppler spectra become close and are determined by the *mss* and not by the orbital velocities. The difference in the shift of the Doppler spectra still remains. Thus, the comparison showed that the proposed approach to developing a semi-empirical model of the Doppler spectrum is effective, provided that the statistical characteristics of the scattering surface are reliably described.

## 3. Results

### 3.1. Comparison of Doppler Spectra

By using an original approach, semi-empirical models of the Doppler spectrum for the ice cover and the sea surface were developed. This allowed us to check the correctness of our assumption about the possibility of using the Doppler spectrum to classify the kind of scattering surface (ice/water) from a moving carrier.

Let us assume that a Ku-band Doppler radar (wavelength 2.1 cm) moves horizontally at a velocity of 200 m/s (see Figure 4). As shown above (see Figure 5), it is sufficient to consider only two variants of the antenna beam: $2° \times 2°$ and $14° \times 2°$. The normalized Doppler spectra for the ice cover (black curve) and sea waves (red curve) for the antenna beam of $2° \times 2°$ (a) and $14° \times 2°$ (b) are shown in Figure 12. It can be seen from the figure that for a narrow antenna beam, the Doppler spectra for sea waves and ice cover practically do not differ. This is in line with the conclusions that follow from Figure 3: surface parameters have little effect on the Doppler spectrum for a narrow antenna pattern, and in this case, the width of the Doppler spectrum is determined by the velocity of the radar.

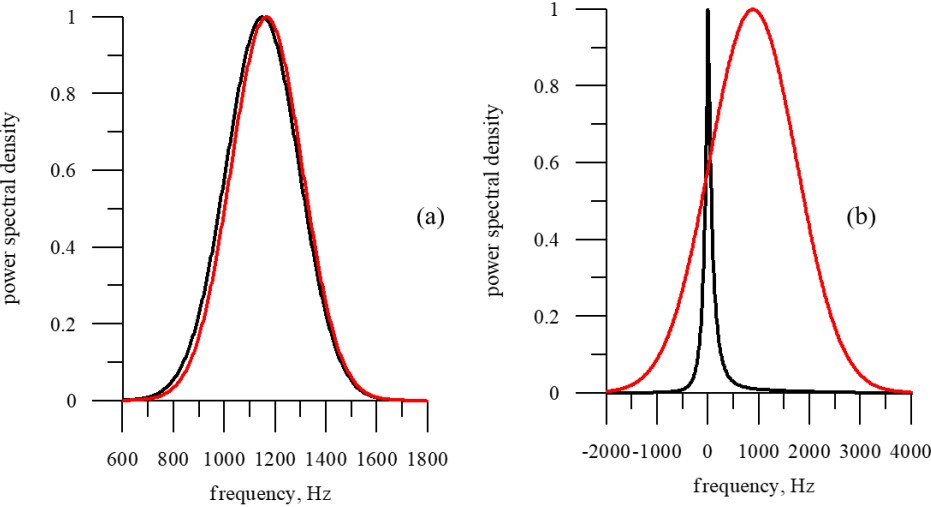

**Figure 12.** Normalized Doppler spectra for a moving carrier ($V_{rad}$ = 200 m/s), incidence angle $\theta_0$ = 5°, azimuth angle = 45°, and two values of the antenna beam, $2° \times 2°$ (**a**) and $14° \times 2°$ (**b**): red curve—sea waves, and black curve—ice cover.

With an increase in the width of the antenna beam, the reflecting surface begins to participate in the formation of the Doppler spectrum, and this is clearly shown in Figure 12b. The "roughness" of the sea surface is much greater than that of the ice cover, so moving radar has a wider Doppler spectrum for the sea surface. The movement of the surface itself in this case does not affect the result. Table 1 shows quantitative estimates of the Doppler spectrum parameters for the considered cases: "sea"—sea waves, and "ice"—ice cover.

**Table 1.** Doppler spectrum parameters for (a) sea waves and (b) ice cover.

| N | $\delta_\alpha$ | $\delta_\beta$ | $f_{shift}$, Hz | $\Delta F_{20}$, Hz | $\Delta F_{42}$, Hz | A | E |
|---|---|---|---|---|---|---|---|
| sea | 2° | 2° | 1166 | 282 | 244 | 0.0 | 0.0 |
| ice | 2° | 2° | 1149 | 290 | 251 | −0.01 | 0.01 |
| sea | 14° | 2° | 887 | 1712 | 1474 | 0.01 | −0.04 |
| ice | 14° | 2° | 100 | 749 | 1733 | 3.5 | 18.4 |

Thus, when using a knife-like antenna beam (14° × 2°) or wide antenna (14° × 14°), there is a significant change in the parameters of the Doppler spectrum during the transition from ice cover to sea waves. Five parameters are changed: two widths and the shift of the Doppler spectrum, and skewness and kurtosis coefficients.

The width and shift of the Doppler spectrum depend on the radar velocity, and, therefore, during measurements, it is necessary to keep the velocity of movement constant. Changing the velocity will change the width and shift of the Doppler spectrum. If one uses the skewness and kurtosis coefficients, this problem is removed, because they characterize the change in the shape of the Doppler spectrum with a change in the kind of underlying surface and do not depend on the velocity. For example, when the velocity changes from 200 m/s to 20 m/s, the shift of the Doppler spectrum for the ice cover changes from about 100 Hz to 10 Hz, while the kurtosis coefficient remains about 18.4.

When calculating the Doppler spectrum, the ice cover scattering diagram for first-year dry (negative air temperature) ice was used. The air temperature (zero crossing) and the type of sea ice will affect the form of the relationship. However, this will affect the absolute values of the RCS, and not the nature of the angular dependence, so the differences in the Doppler spectra measured over the ice cover and the sea surface will remain and all conclusions will remain valid.

*3.2. Influence of Sea Ice Concentration on the Doppler Spectrum*

In the simulation, it was assumed that the reflection comes from the ice cover. The question of the influence of sea ice concentration (SIC) on the Doppler spectrum deserves separate consideration. Even for a sharp "ice–water" boundary, when the carrier crosses it, ice and water in different proportions will fall into the resolution element. When the SIC is less than 1, the reflected signal is the sum of the signals: reflected from the ice cover $RCS_{ice}$ and reflected from sea waves $RCS_{sea}$

$$RCS_{total} = RCS_{ice} \cdot SIC + RCS_{sea}(1 - SIC) \tag{15}$$

where SIC is the sea ice concentration which takes values from 0 (no ice) to 1 (continuous ice cover). As a result, the parameters of the Doppler spectrum will depend on the SIC.

Consider the influence of SIC on the Doppler spectrum of the backscattered radar signal. In the simulation, the SIC is assumed to be uniformly distributed in the footprint, and the radar has a knife-like antenna beam of 14° × 2°.

The results of calculating the Doppler spectrum for different values of SIC are shown in Figure 13. For convenience of comparison, all Doppler spectra were normalized on their own maximum. The Doppler spectrum for SIC = 0 (sea waves) is shown as a solid black curve. The Doppler spectrum for sea waves is the widest and has the largest Doppler shift.

If SIC = 0.5, the Doppler spectrum (blue curve) is formed mainly by the ice cover. With a further increase in SIC, the Doppler spectrum becomes narrower (green curve—SIC = 0.9) and almost coincides with the Doppler spectrum for a solid ice cover, shown as a black dotted line (SIC = 1).

Quantitative estimates of the parameters of the Doppler spectrum for different values of SIC are presented in Table 2.

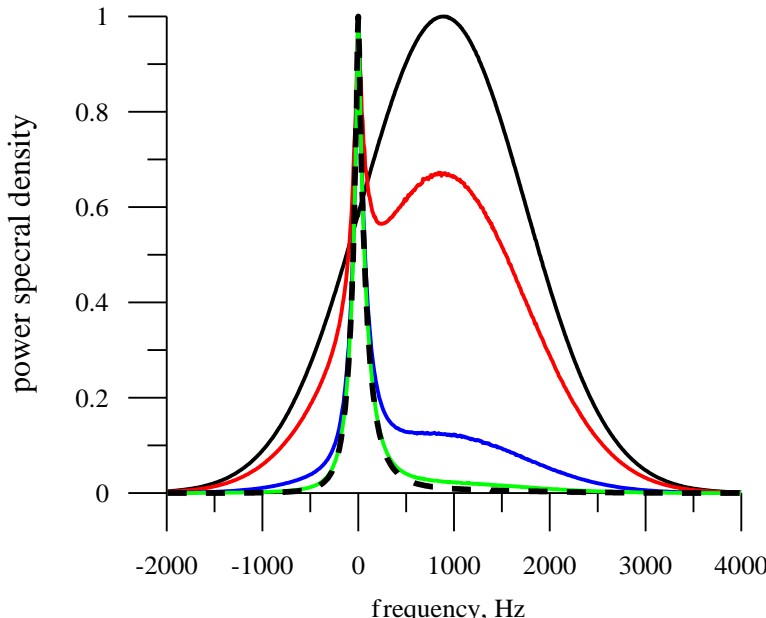

**Figure 13.** Doppler spectra for different SIC: SIC = 0—black curve, SIC = 0.1—red curve, SIC = 0.5—blue curve, SIC = 0.9—green curve, and SIC = 1—black dotted line.

**Table 2.** Doppler spectrum parameters for different SIC.

| SIC | $f_{shift}$, Hz | $\Delta F_{20}$, Hz | $\Delta F_{42}$, Hz | $A$ | $E$ |
|-----|-----|-----|-----|-----|-----|
| 0 | 887 | 1712 | 1474 | 0.01 | −0.04 |
| 0.1 | 825 | 1710 | 1447 | 0.14 | −0.13 |
| 0.5 | 546 | 1584 | 1456 | 0.82 | 0.38 |
| 0.9 | 200 | 1066 | 1684 | 2.36 | 6.99 |
| 1 | 101 | 749 | 1734 | 3.5 | 18.4 |

The table shows that the width and shift of the Doppler spectra are sensitive to changes in the SIC, i.e., it is possible to automate the process of determination of the kind of underlying surface. There is also a change in the coefficients of asymmetry and kurtosis in the transition from the sea surface to the ice cover. The advantage of the skewness and kurtosis coefficients is due to the fact that, in contrast to the width and shift of the Doppler spectrum, they weakly depend on radar velocity.

Table 3 shows the parameters of the Doppler spectrum for a speed of 100 m/s. The width and shift of the Doppler spectrum have become much smaller, and the skewness and kurtosis coefficients are independent of the radar velocity (see Table 2).

**Table 3.** Doppler spectrum parameters for a radar speed of 100 m/s.

| SIC | $f_{shift}$, Hz | $\Delta F_{20}$, Hz | $\Delta F_{42}$, Hz | $A$ | $E$ |
|-----|-----|-----|-----|-----|-----|
| 0 | 443 | 856 | 737 | 0.01 | −0.04 |
| 0.5 | 273 | 792 | 728 | 0.82 | 0.38 |
| 1 | 50 | 375 | 867 | 3.5 | 18.4 |

### 3.3. Impact of Width of Antenna Beam

The width of antenna beam strongly influences the parameters of the Doppler spectrum and determines the efficiency of ice detection. To obtain quantitative estimates, it is necessary to consider the dependence of the Doppler spectrum on the width of the antenna beam. The calculations were made for a radar velocity of 200 m/s, an incidence angle of 5°, a sounding direction of 45°, and four antenna beams: 2° × 2°, 6° × 2°, 10° × 2°, and

$14° \times 2°$. Figure 14 shows Doppler spectra for ice cover (black curve) and sea waves (red curve) for two antenna beams: Figure 14a, $6° \times 2°$, and Figure 14b, $10° \times 2°$.

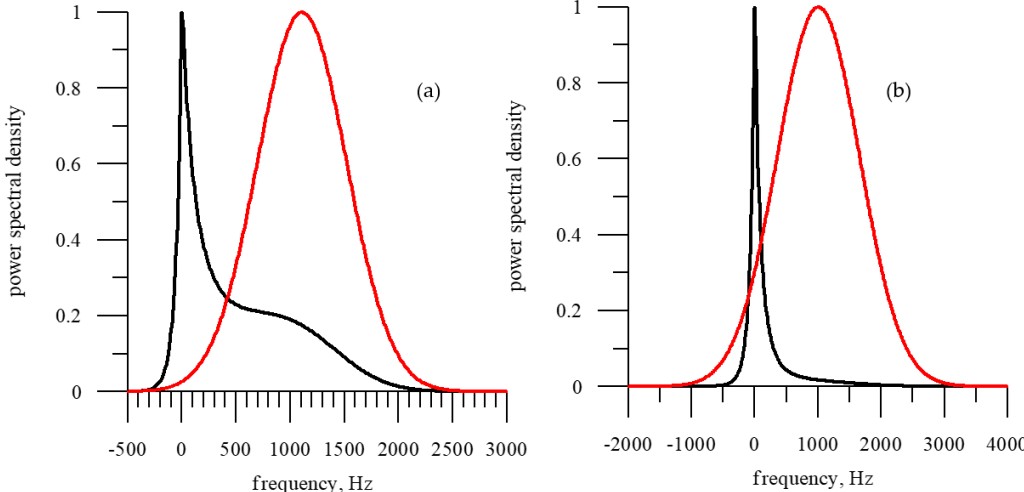

**Figure 14.** Doppler spectra for ice cover (black curve) and sea waves (red curve): (**a**) antenna beam $6° \times 2°$ and (**b**) antenna beam $10° \times 2°$.

It can be seen from the figures that with an increase in the width of the antenna beam from $6°$ (Figure 14a) to $10°$ (Figure 14b), it leads to a significant change in the shape of the Doppler spectrum.

Quantitative estimates are given in Table 4. The wider the antenna beam, the greater the difference between the parameters of the Doppler spectra of ice cover and sea waves. Thus, a knife-like antenna pattern is optimal for detecting the ice cover.

**Table 4.** Doppler spectrum parameters for sea waves (sea) and ice cover (ice) for different antenna beams.

| N | $\delta_\alpha$ | $\delta_\beta$ | $f_{shift}$, **Hz** | $\Delta F_{20}$, **Hz** | $\Delta F_{42}$, **Hz** | *A* | *E* |
|---|---|---|---|---|---|---|---|
| sea | 2° | 2° | 1166 | 282 | 244 | 0.0 | 0.0 |
| ice | 2° | 2° | 1149 | 290 | 251 | −0.01 | 0.01 |
| sea | 6° | 2° | 1107 | 822 | 711 | 0.0 | −0.03 |
| ice | 6° | 2° | 529 | 1048 | 861 | 0.78 | −0.3 |
| sea | 10° | 2° | 1006 | 1305 | 1130 | −0.01 | −0.0 |
| ice | 10° | 2° | 164 | 793 | 1408 | 2.8 | 9.6 |
| sea | 14° | 2° | 887 | 1712 | 1474 | 0.01 | −0.03 |
| ice | 14° | 2° | 100 | 749 | 1733 | 3.5 | 18.4 |

### 3.4. Azimuthal Dependence

In the previous section, the optimal antenna beam was determined. Here, we consider the influence of other parameters of the measurement scheme on the Doppler spectrum: (1) azimuth angle (the angle between the direction of motion and the direction of sounding); (2) incidence angle.

When analyzing the dependence of the Doppler spectrum on the azimuth angle, it is sufficient to take the angle interval from $0°$ to $90°$. The calculations were performed for the initial measurement scheme, and the probing direction (azimuthal angle) was the variable parameter. Figure 15a shows the Doppler spectra for the ice cover Figure 15a and sea waves Figure 15b: black curve—$15°$ azimuth angle, red curve—$45°$, and blue curve—$75°$.

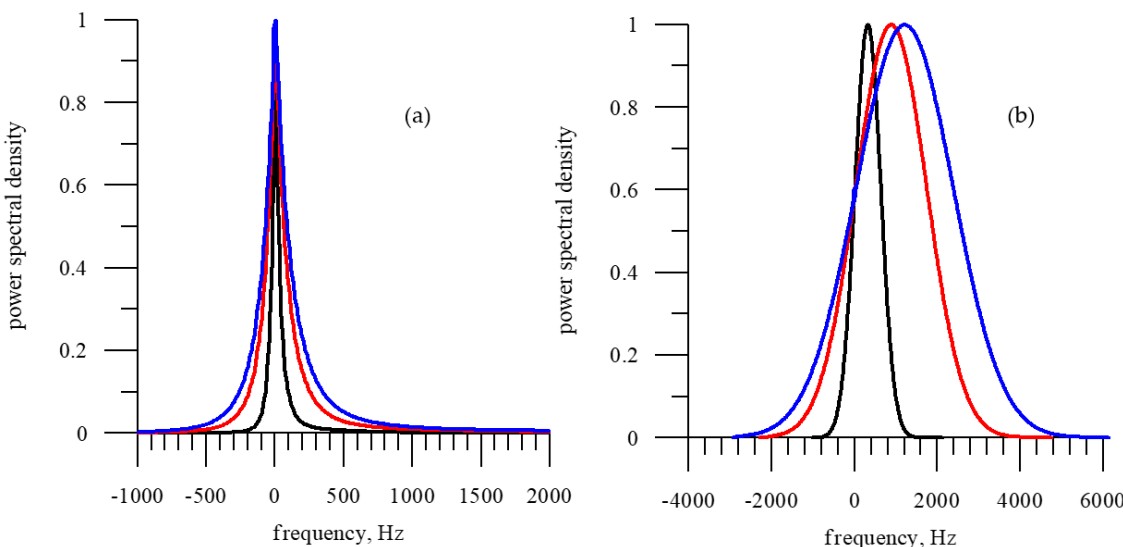

**Figure 15.** Doppler spectra for ice cover (**a**) and sea surface (**b**): black curve—15° azimuth angle, red curve—45°, and blue curve—75°.

As the azimuthal angle increases, the width of the Doppler spectrum increases. For the ice cover, the position of the maximum of the Doppler spectrum does not change and the shift is equal to zero, since the RCS has a maximum at zero incidence angle. For the sea surface, the shift of the Doppler spectrum increases with the increasing azimuth angle. The shift of the Doppler spectrum reaches a maximum at an azimuth angle of 90° when sounding is performed in the direction of motion. Table 5 provides quantitative estimates of the parameters of the Doppler spectrum.

**Table 5.** Doppler spectrum parameters for sea waves (sea) and ice cover (ice) for different azimuth angles.

| N | $\theta_0$ | $f_{shift}$, Hz | $\Delta F_{20}$, Hz | $\Delta F_{42}$, Hz | A | E |
|---|---|---|---|---|---|---|
| sea | 0° | 0 | 37 | 50 | 0.0 | 4.4 |
| ice | 0° | 0 | 12 | 47 | 0.0 | 63.2 |
| sea | 15° | 325 | 628 | 542 | 0.01 | 0.02 |
| ice | 15° | 37 | 29 | 50 | 0.0 | 9.0 |
| sea | 30° | 627 | 1211 | 1043 | 0.01 | −0.03 |
| ice | 30° | 71 | 530 | 1226 | 3.5 | 18.4 |
| sea | 45° | 887 | 1712 | 1474 | 0.01 | −0.04 |
| ice | 45° | 100 | 749 | 1733 | 3.5 | 18.4 |
| sea | 60° | 1086 | 2097 | 1805 | 0.01 | −0.04 |
| ice | 60° | 123 | 918 | 2123 | 3.5 | 18.4 |
| sea | 75° | 1211 | 2338 | 2013 | 0.0 | −0.04 |
| ice | 75° | 137 | 1024 | 2367 | 3.5 | 18.4 |
| sea | 90° | 1254 | 2421 | 2084 | 0.0 | −0.04 |
| ice | 90° | 142 | 1060 | 2451 | 3.5 | 18.4 |

*3.5. Dependence from Incidence Angle*

When calculating, the incidence angle will vary from 0° to 5°. This limitation is due to the fact that the formulas for the RCS are defined for incidence angles of less than 19° (Formulas (7) and (9)). When calculating the Doppler spectrum, integration over the scattering area is performed (see Formula (8)). Integration must be carried out in infinite limits; however, in the numerical calculations, the integration limits are determined by the incidence angle, at which the power of the reflected signal is close to zero, and with a further increase in the integration limits, the RCS does not change.

Estimates have shown that for a wide antenna beam (14°), it is sufficient to use limits of +/−14°. In this case, the error associated with controlling the limits of integration is less than one percent, and this will not affect the correctness of the conclusions. Therefore, the maximum incidence angle equals 5°. The simulation was performed for the same conditions: radar velocity 200 m/s, sounding direction 45°, antenna beam 14° × 2°, and incidence angles 1° (black curve), 3° (blue curve), and 5° (red curve). Three variants were considered: ice cover, sea waves, and SIC at the level of 50%. The Doppler spectra for all three cases of the scattering surface, a—ice, b—sea waves, and c—SIC = 0.5, are shown in Figure 16.

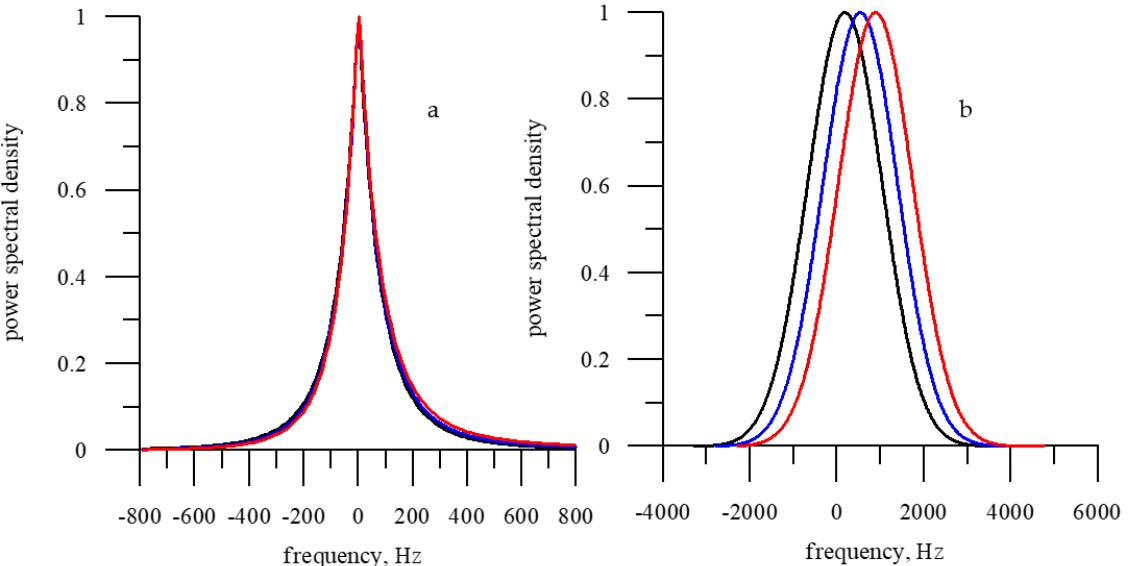

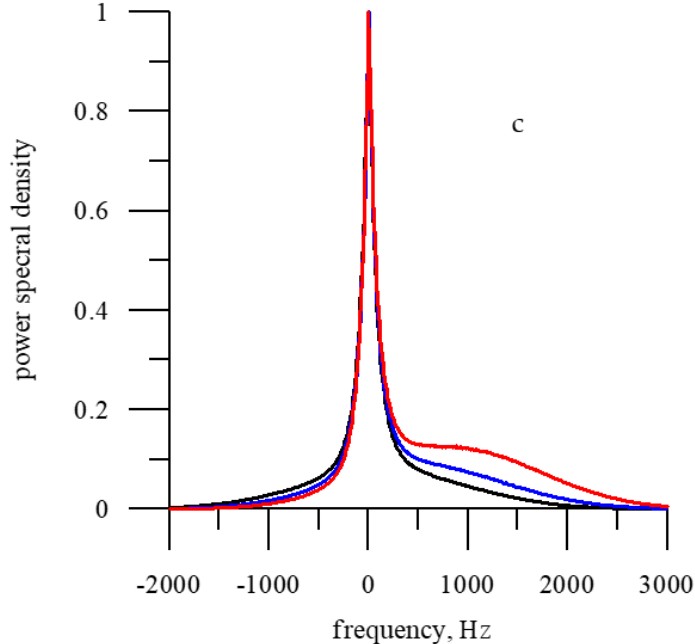

**Figure 16.** Doppler spectra for ice cover (**a**), sea waves (**b**), and SIC = 0.5 (**c**). Radar velocity 200 m/s, sounding direction 45°, antenna beam 14° × 2°, and incidence angles 1° (black curve), 3° (blue curve), and 5° (red curve).

Changing the incidence angle has almost no effect on the shift of the Doppler spectrum for the ice cover (see Figure 16a). This is explained by the fact that the maximum power of the backscattered signal corresponds to a zero incidence angle.

For the sea waves, with an increase in the incidence angle, an increase in the Doppler spectrum shift is observed (see Figure 16b) while maintaining all other parameters, i.e., by taking measurements at different incidence angles, it is possible to determine the kind of scattering surface. To achieve this, it is sufficient to observe how the parameters of the Doppler spectrum change with a change of incidence angle.

For an SIC other than zero, there is a lack of determining the Doppler spectrum shift through the center of gravity of the spectrum (Formula (10)). The position of the maximum of the Doppler spectrum does not change; however, due to a change in the shape of the spectrum, the Doppler spectrum shift changes (see Figure 16c). Therefore, this parameter can be additionally used in the analysis to describe the Doppler spectrum.

The exact values are given in Table 6.

**Table 6.** Doppler spectrum parameters for sea waves (sea), ice cover (ice), and SIC = 0.5 for different incidence angles.

| N | $\theta_0$ | $f_{shift}$, Hz | $\Delta F_{20}$, Hz | $\Delta F_{42}$, Hz | A | E |
|---|---|---|---|---|---|---|
| Ice | 0° | 0 | 496 | 1166 | 0.0 | 19.1 |
| Ice | 1° | 14 | 504 | 1198 | 1.1 | 19.5 |
| Ice | 2° | 30 | 531 | 1289 | 2.1 | 20.6 |
| Ice | 3° | 49 | 577 | 1421 | 2.9 | 21.3 |
| Ice | 4° | 71 | 648 | 1576 | 3.4 | 20.6 |
| Ice | 5° | 100 | 749 | 1733 | 3.5 | 18.4 |
| Sea | 0° | 0 | 1716 | 1479 | 0.0 | 0.0 |
| Sea | 1° | 179 | 1715 | 1478 | 0.0 | −0.03 |
| Sea | 2° | 356 | 1714 | 1477 | 0.0 | −0.03 |
| Sea | 3° | 533 | 1713 | 1476 | 0.0 | −0.03 |
| Sea | 4° | 710 | 1713 | 1475 | 0.0 | −0.03 |
| Sea | 5° | 887 | 1712 | 1474 | 0.0 | −0.03 |
| Sic50 | 0° | 0 | 1204 | 1453 | 0.0 | 2.8 |
| Sic50 | 1° | 89 | 1222 | 1454 | 0.4 | 2.7 |
| Sic50 | 2° | 182 | 1275 | 1458 | 0.7 | 2.2 |
| Sic50 | 3° | 286 | 1357 | 1463 | 0.9 | 1.6 |
| Sic50 | 4° | 406 | 1464 | 1462 | 0.9 | 1.0 |
| Sic50 | 5° | 546 | 1584 | 1456 | 0.8 | 0.4 |

## 4. Discussion

For the first time, a theoretical study was conducted of the properties of the Doppler spectrum of a microwave radar signal measured while the radar was moving over an ice cover. To develop a semi-empirical model of the Doppler spectrum, an original approach was used, which is based on the scattering diagram of the ice cover. To calculate the dependence of the RCS on the incidence angle, we used data from the Ku-band orbital precipitation radar of the GPM satellite, which takes measurements at low incidence angles (<19°). As a result of the regression analysis, a formula was obtained for the dependence of the RCS on the incidence angle for the ice cover (first-year ice at negative air temperature).

To evaluate the effectiveness of the proposed approach to developing a semi-empirical model of the Doppler spectrum, it was applied for sea waves, for which a theoretical model of the Doppler spectrum exists. The comparison of both models confirmed the correctness of the new approach and, consequently, the correctness of the developed model of the Doppler spectrum for the ice cover.

In the course of the study, it was shown that at small incidence angles, the Doppler spectra for ice and sea waves differ significantly, and the kind of underlying surface can be determined from the Doppler spectrum. The problem is solved for continuous ice cover and sea waves when SIC = 1 and SIC = 0, respectively.

The case when the SIC is not equal to 1 was considered separately. With a formal approach to the analysis of the Doppler spectrum, one can determine not only the kind of scattering surface (ice/water) but also the SIC using Formulas (7) and (9). However, it will not be easy to realize this in practice. The fact is that the formulas used describe the average dependences for the ice cover and sea waves. For a specific wind-wave situation in the study area, these dependences may differ significantly from the real ones and, therefore, the assessment of the ice cover concentration will be incorrect. Figure 17 provides an example of the dependences of the RCS on the incidence angle for two wind speeds: dotted curves represent 12 m/s, and solid curves represent 4 m/s. The calculations were performed according to the theoretical model (see Formula (14)).

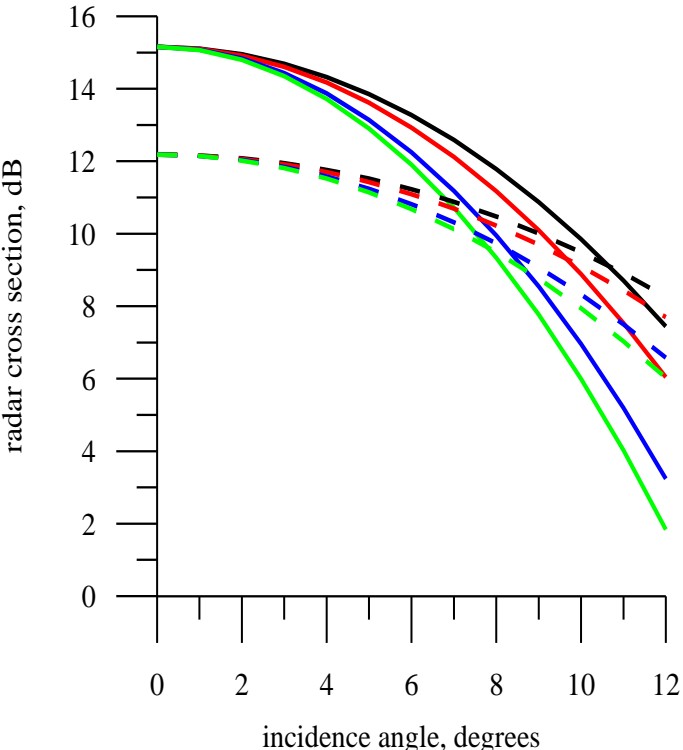

**Figure 17.** Dependence of the RCS on the incidence angle for wind speeds of 12 m/s (dashed curve) and 4 m/s (solid curve) for different azimuth angles: black curve—0°, blue curve—30°, red curve—60°, and green curve—90°.

In addition to wind speed, the direction of wave propagation also affects the RCS. The dependence of the RCS on the incidence angle is sensitive to the direction of wave propagation, and the curves show the angular dependences for four azimuth angles (the angles between the sounding direction and the direction of wave propagation): black curve—0°, blue curve—30°, red curve—60°, and green curve—90°. Therefore, depending on the wind speed and direction, different estimates of the SIC will be obtained.

For the ice cover, the RCS depends on the type of ice and the air temperature, which also makes the problem of estimating the SIC ambiguous. Nevertheless, the problem of determining the SIC has a solution if we use local, rather than universal, dependencies. To calculate the local dependences of the RCS on incidence angle, it is necessary to use two reference points, solid ice cover (SIC = 1) and sea waves (SIC = 0), for the study area. They can be calculated directly during the measurement, when the radar moves, for example, from an area of continuous ice cover (SIC = 1) to an area of open water (SIC = 0).

In this case, the dependences of the RCS on the incidence angle for the ice cover and the sea waves will be calculated. The use of "local" (measured) dependencies will allow determining the SIC in the investigated area. The proposed approach to developing a semi-empirical model of the Doppler spectrum, which was used for the area of small incidence

angles, is universal and can be applied across the entire interval of incidence angles. The choice of the small incidence angles in this study is due to the fact that precipitation radar data, which take measurements at incidence angles of less than 19°, were used.

In the transition from ice cover to sea waves, there is a significant change in the width and shift of the Doppler spectrum. However, these parameters are sensitive to radar velocity, which complicates the task, because it imposes requirements on the constancy of the movement velocity. Numerical modeling has shown that the use of additional parameters to describe the Doppler spectrum, skewness, and kurtosis coefficients makes it possible to remove the dependence of the result on radar velocity.

It was shown that the problem of determining the kind of underlying surface can only be solved for a radar with a wide or knife-like (by incidence angle) antenna beam. Only in this case does the radar begin to "see" the scattering surface. The need to use a wide (knife-like) antenna beam limits the possibility of using this approach for orbital radars due to the size of the scattering area (footprint). It is necessary to apply a special measurement scheme that will improve the resolution.

In this study, the influence of the main characteristics of the measurement scheme on the Doppler spectrum was considered: (1) the incidence angle; (2) the velocity of the radar movement; (3) the direction of probing. When studying the RCS dependence on the incidence angle, a small interval of incidence angles was considered. The choice of the interval of incidence angles (0°–5°) is due to the fact that the regression dependences used for the RCS were obtained for incidence angles of 0°–19°. With formulas for RCS that are valid over a wider range of incidence angles, the semi-empirical model can be extended to a larger interval of incidence angles. Numerical modeling has shown that as the incidence angle increases, the Doppler spectrum becomes more sensitive to the kind of scattering surface, provided that a knife-like or wide antenna beam is used.

The next measurement scheme parameter that can be changed is the sounding direction or azimuth angle. Modeling has shown that, at any azimuth angle, a significant difference remains between the Doppler spectra of the sea surface and the ice cover.

Another important parameter that affects the parameters of the Doppler spectrum is the width of the antenna beam. During the simulation, the width of antenna beam varied from 2° to 14°. The limitation on the maximum width of the antenna beam is also related to the experimental data at our disposal. However, such an interval of incidence angles is sufficient to answer the question of choosing the antenna beam. The wider the antenna, the greater the difference between the Doppler spectra.

## 5. Conclusions

For the first time, a semi-empirical model of the Doppler spectrum of a radar signal backscattered by an ice cover at small incidence angles has been developed. The inclusion of the antenna beamwidth in the model made it possible to consider various measurement schemes. In the course of the study, the main factors affecting the parameters of the Doppler spectrum of the backscattered radar signal when measured from a moving carrier were considered. It has been shown that to determine the kind of underlying surface, it is necessary to use a knife-like antenna beam.

Thus, as a result of the research, it was shown that the Doppler spectrum is an effective tool for determining the kind of underlying surface according to the "ice–open water" criterion. When using local dependences of the RCS for solid ice cover and sea waves (two reference points), it will be possible to estimate the SIC from the Doppler spectrum.

The obtained conclusions about the prospects of using the Doppler spectrum for classifying the kind of the scattering surface are based on the developed semi-empirical model of the Doppler spectrum for the ice cover. In February 2022, an experiment was conducted to verify the results. For the first time, the Doppler spectrum was measured from a moving carrier during nadir sounding by a radar with a knife-like antenna beam (4 × 30). The radar (X-band) was installed on the technological trolley of the Nizhny Novgorod cable car, which crosses the Volga River.

The data are currently being processed and the results will be published in the next paper.

**Author Contributions:** V.K., conceptualization and writing; Y.T., numerical simulation; M.P., writing—original draft preparation; M.R., visualization; E.M. initiated the study; K.P., software and numerical simulation. All authors have read and agreed to the published version of the manuscript.

**Funding:** This research was funded by the Russian Foundation for Basic Research (project no. 20-05-00462a) and State Task of IAP RAS N 0030-2021-0006.

**Data Availability Statement:** Not applicable.

**Acknowledgments:** The DPR data are presented by the JAXA (JAXA Satellite Project Research (Non-Funded) PI N ER3GPN103).

**Conflicts of Interest:** The authors declare no conflict of interest.

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
