# Peer review of "Application of the Doppler Spectrum of the Backscattering Microwave Signal for Monitoring of Ice Cover: A Theoretical View"

_remotesensing, doi:10.3390/rs14102331_

Round 1

Reviewer 1 Report

paper is good and has few errors  - see attached

Author Response

Please, find reply inthe attached file.

Reviewer 2 Report

This revised version is much easier to read, and has addressed a number of changes from my own review, as well as others.  My only comment at this point would be to consider a wider reach of literature as there are a number of Canadian author names I don't see in the reference list that have produced work that work strongly support this paper further. 

That said, I'll leave it up to the authors.

Author Response

please, find reply at the attached file.

Reviewer 3 Report

The writing and organization of this work are good. Both Abstract and conclusions give a clear idea about this work. In this study, the authors discuss the possibility of using the Doppler spectrum of the reflected microwave signal to overcome the errors in estimating the area of the ice cover.  The presented results show that the Doppler spectrum is an effective tool for determining the kind of underlying surface.  I don’t have any specific comments as the authors have presented results and discussion nicely. I have found a few issues which can improve the quality of this work.

  1. The quality of Figure 1 must be improved. Figure labeling is not fully clear.
  2. Figures 1(a) and 1(b) must be separately discussed for better understanding of readers. Authors must refer to each figure in the discussion.
  3. Authors should provide a precise description of Figure 2 and additional discussion must be added into the manuscript body.
  4. Figure 2 must be labeled properly for better clarity. What is on X-axis and Y-axis?
  5. The quality of Figure 6 is not good. Figures with high resolution must be provided for better readability.
  6. Would you like to confirm Axis Y in line 126, is it a typo in the writing format of Y?
  7. Figure 3. Make relevant corrections in the X-axis label. (W) width of the antenna beam. (,) degrees
  8. Line 202, height of H0 (H0)
  9. Check Figure 4 and make relevant corrections in the variables writing format.
  10. Authors should carefully revise the figure label and make suitable writing changes.
  11. Line 532 and 537: is it Figure 10(a) or 11(a) as I cannot find Figure 10(a).
  12. Author should provide discussion of both 14(a) and 14(b) separately. Similarly, Figures 16(a), 16(b), and 16(c) must be discussed.
  13. The experiments results are reasonable and interesting.
  14. Although the significant effort made by the author mainly in the collection of information, the writing still needs to be improved and avoid certain writing formats and some minor corrections. 
  15. Authors should highlight some possible future research directions, particularly with regard to helping relevant research fraternity for further contributions.
  16. It is suggested to change Section 4 into Discussion and add another Section Conclusion to precisely introduce about research contributions of this study.
  17. Reference section is updated with very recent research contributions. I have found some interesting studies. However, there is some inconsistency in reference writing. Authors should refer to the journal template.

In my opinion, this article is based on interesting research contributions. However, minor changes are still required to further improve presentation quality and re-organize some contents.

Author Response

please, find the reply in the attached file.

This manuscript is a resubmission of an earlier submission. The following is a list of the peer review reports and author responses from that submission.

Round 1

Reviewer 1 Report

Overall, I think the work presented in this paper is very impressive, and well-described methods that can improve our use of radar backscatter using the Doppler spectrum of the reflected microwave signal to observe sea ice. 

Introduction is a bit too short for my liking.  I appreciate the desire to get to your work, but a bit more background / literature review of similar work, or work to build on is requested. 

Line 83 when electromagnetic waves "are" reflected -- check use of 'is/are' tenses throughout manuscript.  

There are a number of very short paragraphs in this manuscript, some of which consist of only one sentence.   Could these be assembled into more complete paragraphs as flowing prose?

Please focus on revising your manuscript into better form with an expanded introduction, english style / grammar checking, and construction of fuller paragraphs to describe your results.  The work and results themselves are good, and need no further analysis in my opinion. 

Reviewer 2 Report

To distinguish the sea ice from sea waves with backscattering radar cross, this paper gave a semi-empirical model of the Doppler spectrum of sea ice and a few parameters, including shift, width and skewness, were used to describe the Doppler spectrum. The influence characters of Doppler spectrum were studied, including movement velocity, antenna beam width, incidence angle, and SIC. But this munuscript needs to be modified further.

1, The whole MS looks like a log file of a research. The structure of the MS should be reorganized: introduction, data, method(initial assumption, semi-empirical model, basic parameters of Doppler spectrum), result and discussion (section 6-11), conclusion.

2, The GPM satellite data were used for the semi-empirical. In line 221, TRMM and GPM data were used. The detailed information of satellite data and research area should be given.

3, The semi-empirical model of sea ice is based on the observation of first year ice. How about Multi year ice?

4, The conclusion should be modified further.

5, According to the model, several characters’ influence of Doppler spectrum is considered. Which one should be considered first when distinguishing sea ice and water.

Reviewer 3 Report

Extensive editing required - paper is hard to understand
